# Development of the CRISPR-Cas9 System for the Marine-Derived Fungi *Spiromastix* sp. SCSIO F190 and *Aspergillus* sp. SCSIO SX7S7

**DOI:** 10.3390/jof8070715

**Published:** 2022-07-08

**Authors:** Yingying Chen, Cunlei Cai, Jiafan Yang, Junjie Shi, Yongxiang Song, Dan Hu, Junying Ma, Jianhua Ju

**Affiliations:** 1CAS Key Laboratory of Tropical Marine Bio-Resources and Ecology, Guangdong Key Laboratory of Marine Materia Medica, RNAM Center for Marine Microbiology, South China Sea Institute of Oceanology, Chinese Academy of Sciences, 164 West Xingang Road, Guangzhou 510301, China; yychen@scsio.ac.cn (Y.C.); 18262628958@163.com (C.C.); 15050316090@163.com (J.Y.); shijunjie21@mails.ucas.ac.cn (J.S.); songx@scsio.ac.cn (Y.S.); 2Southern Marine Science and Engineering Guangdong Laboratory (Guangzhou), No. 1119, Haibin Road, Nansha District, Guangzhou 511458, China; 3College of Oceanology, University of Chinese Academy of Sciences, Qingdao 266400, China; 4Guangdong Province Key Laboratory of Pharmacodynamic Constituents of TCM and New Drugs Research, Institute of Traditional Chinese Medicine and Natural Products, College of Pharmacy, Jinan University, Guangzhou 510632, China; thudan@jnu.edu.cn

**Keywords:** marine-derived fungi, CRISPR-Cas9, *Spiromastix* sp. SCSIO F190, *Aspergillus* sp. SCSIO SX7S7, secondary metabolites, histone deacetylase

## Abstract

Marine-derived fungi are emerging as attractive producers of structurally novel secondary metabolites with diverse bioactivities. However, the lack of efficient genetic tools limits the discovery of novel compounds and the elucidation of biosynthesis mechanisms. Here, we firstly established an effective PEG-mediated chemical transformation system for protoplasts in two marine-derived fungi, *Spiromastix* sp. SCSIO F190 and *Aspergillus* sp. SCSIO SX7S7. Next, we developed a simple and versatile CRISPR-Cas9-based gene disruption strategy by transforming a target fungus with a single plasmid. We found that the transformation with a circular plasmid encoding *cas9*, a single-guide RNA (sgRNA), and a selectable marker resulted in a high frequency of targeted and insertional gene mutations in both marine-derived fungal strains. In addition, the histone deacetylase gene *rpd3* was mutated using the established CRISPR-Cas9 system, thereby activating novel secondary metabolites that were not produced in the wild-type strain. Taken together, a versatile CRISPR-Cas9-based gene disruption method was established, which will promote the discovery of novel natural products and further biological studies.

## 1. Introduction

Marine fungi are important sources of novel natural products for drug discovery due to their unique defenses that allow them to survive under extreme conditions. The extreme sea environments are typically characterized by the absence of sunlight, high hydrostatic pressure, oligotrophy, and low temperature, which can induce extremophilic fungi to produce secondary metabolites with diverse biological activities [1,2]. Cephalosporin C, a *β*-lactam-type natural antibiotic, was discovered from a *Cephalosporium* species isolated from the Sardinian coast, representing the first fungal antibiotic isolated from a marine environment [3]. Until 2021, Gomes and his colleagues cataloged antimicrobials from marine fungi and reported 108 compounds with antibacterial potential that can be developed to new drug leads [4]. Additionally, the secondary metabolites from marine fungi also hold promise as antitumor and anti-inflammatory therapeutic agents [1,5]. With the advent of the post-genomic era, the surge of genomic data available suggests that the biosynthetic potential of fungi is more immense than previously anticipated. However, most of the biosynthetic gene clusters are either dormant or expressed at very low levels under laboratory growth conditions. The genetic engineering of the largely unexplored fungal species (e.g., marine-derived fungi) would be an efficient strategy to induce the expression of silent gene clusters that could lead to the discovery of novel bioactive compounds [6,7].

Although the genetic manipulation systems in filamentous fungi are being explored, the deeper exploration of important pharmaceutical secondary metabolites has been hampered by a lack of versatile, simple, and highly efficient genetic manipulation techniques [8]. Fortunately, the emergence of CRISPR (clustered regularly interspaced short palindromic repeats)-Cas9-based genome editing technology has revolutionized genetic research and has raised hopes for discovering new bioactive compounds from fungi. The CRISPR-Cas9 system consists of two components, a single chimeric guide RNA (sgRNA) and a Cas9 nuclease. The Cas9 nuclease is guided by sgRNA, introducing specific DNA double-strand breaks (DSBs) that are located directly 3 bp upstream of a protospacer-adjacent motif (PAM) [9]. The break is then repaired by the non-homologous end-joining (NHEJ) pathway, resulting in deletions and substitutions, thereby inactivating the target gene [10]. To date, this technology has been widely applied across species, including various fungal strains, such as *Trichoderma reesei* [11], *Pyricularia oryzae* [12], *Aspergillus oryzae* [13], *Aspergillus niger* [14], *Aspergillus fumigatus* [15], *Aspergillus nidulans* [16], *Talaromyces atroroseus* [17], *Neurospora crassa* [18], *Candida albicans* [19], *Alternaria alternata* [20], *Penicillium chrysogenum* [21], *Ustilago maydis* [22], *Acremonium chrysogenum* [23], *Glarea lozoyensis* [24], *Nodulisporium* sp., and *Sporormiella minima* [25]. However, the applications of the CRISPR-Cas9 techniques are still limited to a small subset of genetically tractable Ascomycota that has been relatively thoroughly studied. Therefore, developing species-specific and efficient CRISPR-Cas9-based methods is crucial for the genetic manipulation and exploitation of novel natural products of fungi isolated from various environments.

Both *Spiromastix* sp. SCSIO F190 and *Aspergillus* sp. SCSIO SX7S7 are marine-derived fungi. We have isolated several antibacterial and antifungal compounds from these two strains [26,27]. In addition to the natural products expressed under normal conditions, genome sequencing and antiSMASH analyses revealed that both strains possess various cryptic secondary metabolites that can be activated. Moreover, *Spiromastix* sp. SCSIO F190 is also an excellent model for studying the interactions of marine fungi and endo-bacteria [26]. Therefore, establishing a reproducible and efficient genetic manipulation system is vital for studying the biosynthesis pathways of isolated compounds, activating silent secondary metabolite gene clusters, and revealing the molecular interactions between the marine fungal host and endo-bacteria.

In this study, we develop all of the critical steps for transforming and establishing a stable and reproducible CRISPR-Cas9-based genetic disruption system for the two marine-derived fungi. Next, we successfully use it to delete the regulatory genes *creA* and *cak1* in *Spiromastix* sp. SCSIO F190 and *Aspergillus* sp. SCSIO SX7S7, respectively. Moreover, the histone-deacetylase-encoding gene *rpd3* is also disrupted, leading to the activation of various novel secondary metabolites.

## 2. Materials and Methods

### 2.1. Strains, Plasmids, and Culture Conditions

*Spiromastix* sp. SCSIO F190 was obtained from a marine sediment sample collected from the Northern South China Sea, while *Aspergillus* sp. SCSIO SX7S7 is a coral-derived epiphytic fungus that was obtained from the South China Sea. Both strains were routinely maintained on potato dextrose agar (PDA) or in potato dextrose broth (PDB) at 28 °C. A complete medium [28] was used for the sporulation and PDB was used for the protoplast preparation. An antibiotic sensitivity test was carried out on *Aspergillus* nitrogen-free minimal (ANM) medium [28] containing corresponding antibiotics. *Escherichia coli* strain DH5α was cultured in Luria–Bertani (LB) medium with appropriate antibiotics for plasmid DNA isolation. A plasmid containing *Trichoderma reesei* codon-optimized *cas9* (to*Cas9*) and hygromycin B phosphotransferase (*hph*) coding sequences was kindly provided by Prof. Dan Hu from Jinan University [25]. A plasmid pFC330 containing the sgRNA expression sequence was generously offered by Prof. Yi Tang from the University of California, Los Angeles, CA, USA. These two vectors were used to construct the *cas9* and sgRNA-expressing vector pBSKII-to*Cas9*-*hph*-sgRNA used in this study. All strains and plasmids used in this study are listed in Appendix A.

### 2.2. DNA Manipulation

The primer synthesis and DNA sequencing were performed by Sangon Biotech Co., Ltd. (Shanghai, China) and Qingke Biotech Co., Ltd. (Guangzhou, China) (Appendix A). The plasmid extraction and DNA purification were carried out with commercial kits (Omega Bio-Tech, Inc., Norcorss, GA, USA). The restriction enzymes and other DNA modification reagents were purchased from New England Biolabs, Inc. (NEB, Hitchin, UK) and Takara Biotechnology Co., Ltd. (Dalian, China). The DNA amplification was carried out on a PCR thermal cycler from Applied Biosystems (Thermo Fisher, Shanghai, China) with either Taq DNA polymerase (TaKaRa, Dalian, China) or High-Fidelity Fastpfu DNA polymerase (TransGen, Beijing, China). The isothermal assembly of multiple DNA fragments was performed using a pEASY^®^-Uni Seamless Cloning and Assembly Kit from TransGen Biotechnology Co., Ltd. (Beijing, China).

### 2.3. Maker Gene Selection and Fungal Sensitivity Test

The antibiotic sensitivity levels were determined in ANM medium with varying concentrations of hygromycin B, geneticin, pyrithiamine, zeocin, and glufosinate. Concentrations of 200 and 400 μg mL^−1^ were applied, except for pyrithiamine (2 and 10 μg mL^−1^) and geneticin (50 and 200 μg mL^−1^). Control plates without antibiotics were used. The strains were point-inoculated and grown on solid ANM media with and without antibiotics at 28 °C for 4–8 days.

### 2.4. Construction of Cas9 and gRNA Expression Vectors

CRISPR-Cas9 vectors with specific sgRNA genes, harboring the respective protospacer and a 6 bp inverted repeat of the 5′-end of the protospacer to complete the hammer-head cleavage site, were generated in a single isothermal assembly step. Firstly, pBSKII-to*Cas9*-*hph* was cut by the BsaAI restriction enzyme. Then, sgRNA together with ribozyme sequences controlled by the P*gdpA* and T*trpC* sequences was separately amplified from pFC330 using two primer pairs Frag1-F/R and Frag2-F/R. The primers of Frag1-F and Frag2-R are universal primers for all genes carrying 5′-end sequences complementary to the ends of the plasmid pBSKII-to*Cas9*-*hph* cut with BsaAI, as outlined in Appendix A. Frag1-R is a gene-specific primer containing a 5′-end hammerhead (HH) ribozyme sequence and a 6 bp inverted repeat of the 5′-end of the protospacer. Frag2-F is also a gene-specific primer containing a gene-specific protospacer, partial sgRNA backbone sequence, and 5′-end sequences complementary to the end of Fragment1. The two fragments amplified by Frag1-F/R and Frag2-F/R are inserted into the BsaAI-digested pBSKII-to*Cas9*-*hph* plasmid by the isothermal assembly.

### 2.5. Optimization of Protoplast Preparation

In order to examine the effect of the enzyme digestion on the protoplast yields of different fungal ages, a total of 1 × 10^9^ conidia were cultured in 50 mL potato dextrose broth (PDB) at 100 rpm and 28 °C for 12, 28, 24, and 36 h. After completion of the culture, the mycelia were collected via centrifugation at 5000 rpm for 10 min. The collected hyphae were washed twice with sterile 0.6 M MgSO_4_ and then digested with different concentrations of lysing enzymes (Sigma-Aldrich, St. Louis, MO, USA) and Yatalase (TaKaRa, Dalian, China) for different times in a digestive solution with 1.2 M MgSO_4_ and 10 mM sodium phosphate buffer at pH 7.0. The digestive solution and the hyphae were mixed on a shaker at 100 rpm at 28 °C. The digested solutions were checked under a microscope for the protoplast morphology and counting. The first microscopic observation was at 2 h, then at 30 min intervals until homogeneous protoplasts, about twice the size of spores, were formed. The protoplast suspension was transferred into a transparent centrifuge tube and gently overlaid with an equal volume of trapping buffer (0.6 M sorbitol, 100 mM Tris/HCl, pH = 7.0), then spun for 30 min at 4000 rpm (4000× *g*) at 4 °C. The mycelial debris pelleted and a white band of protoplasts formed at the interface. Protoplast bands were taken from the interface with a Pasteur pipette and pooled into the third chilled Falcon tube. Then, an equal volume of STC (0.01 M Tris/HCl pH 7.5, 0.01 M CaCl_2_, and 1.2 M sorbitol) buffer was added into the Falcon tube and the protoplast was precipitated via centrifugation and washed once again with 10 mL STC. Finally, the protoplasts were resuspended in STC solution to a concentration of 2 × 10^7^ mL^−1^ for the subsequent transformation.

### 2.6. Transformation and Regeneration of Protoplasts

About 2–5 μg of *cas9* and gRNA expression plasmid was added into 100 μL of the above protoplast suspension, then 25 μL of 60% PEG was added to each tube and mixed gently. Next, 1 mL of 60% polyethylene glycol (PEG) was added to the DNA–protoplast mixture after 30 min incubation on ice with gentle mixing. The PEG-treated protoplast suspension was diluted with 5 mL STC solution and centrifuged at 2000× *g* for 10 min at 4 °C. Next, the protoplasts were resuspended in 200 μL of STC solution and poured onto 3–4 regeneration plates (ANM with 1.2 M sorbitol) containing 1.5% agar and hygromycin B. After 4–14 days cultivation at 28 °C, the transformants were inoculated onto a new PDA plate (without sorbitol).

### 2.7. Transformant Validation via MSBSP-PCR and Sequencing

The transformants were randomly picked and cultured in PDA for genomic DNA extraction. The first PCR reaction was performed using genomic DNA and a primer designed to anneal at the recognition sites of the gRNA. Based on the principle of the site-based specific primers polymerase chain reaction (MSBSP-PCR) described by Guo et al. [29], the first-round screening would identify CRISPR-Cas9-induced mutations located close to the DSB site that occur 3 bp upstream of the PAM. The second PCR reaction was carried out using genomic DNA and primers located upstream and downstream of the target gene, which would detect gene disruptions situated far from the PAM. The strains that either failed to show positive PCR bands or with obvious phenotypes were further validated via the sequencing of the PCR products amplified with primer pairs flanking the target gene.

### 2.8. Secondary Metabolite Analysis

To examine the production of secondary metabolites, both mutants and wild-type strains were cultivated in PDB liquid medium for 7 days at 28 °C, 180 rpm. The cultures were extracted with butanone, the organic phases were evaporated to dryness, and the remaining residues were redissolved in 1 mL MeOH, 30 μL of which was injected into the analytical HPLC for analysis. The HPLC analysis was performed on an Agilent 1260 HPLC system (Agilent Technologies Inc., Santa Clara, CA, USA) equipped with a binary pump and a diode array detector using an analytical Phenomenex column (250 × 4.60 mm, 5 microns). The samples were eluted with a linear gradient of 0% to 80% solvent B over 20 min, followed by 70% to 100% solvent B in 1.5 min, and then eluted with 100% solvent B in 5.5 min at a flow rate of 1.0 mL/min using UV detection at 220 nm, 254 nm, 275 nm, and 354 nm.

## 3. Results

### 3.1. Antibiotic Sensitivity Test and Resistance Maker Gene Selection

The establishment of an effective selection system for differentiating transformed isolates from untransformed ones is a key step in genetic tool development. In this study, we tested the fungal sensitivity to five commonly used antibiotics with available resistance genes: hygromycin B, zeocin, pyrithiamine, glufosinate, and G418. Fresh spores of *Spiromastix* sp. SCSIO F190 and *Aspergillus* sp. SCSIO SX7S7 strains were pointed and inoculated in ANM plates with two concentrations of the above antibiotics. After 4 or 8 days of culture, *Aspergillus* sp. SCSIO SX7S7 was not able to grow in the plate containing hygromycin, zeocin, and pyrithiamine, but only hygromycin B could inhibit the growth of *Spiromastix* sp. SCSIO F190 (Figure 1). Therefore, the resistance gene hygromycin B phosphotransferase (*hph*) was chosen as the selection marker gene for both strains.

### 3.2. Establishment of Protoplast Preparation and Transformation

A viable and stable protoplast formation system is the basis for protoplast-based transformations. The first step in the protoplast preparation is removing the cell wall through enzymatic digestion. The fungal cell wall structure is highly variable among different species and highly dynamic during the growth of fungi, including during spore germination, hyphal branching, and the formation of the diaphragm [30]. Thus, species-specific transformation protocols must be optimized for each strain, especially for those seldom-studied wild-type marine fungi. Here, two cell wall digestion enzymes, lysing enzyme and Yatalase, were combined to test the protoplast release from *Spiromastix* sp. SCSIO F190 and *Aspergillus* sp. SCSIO SX7S7. The effects of the fungal ages and enzyme incubation time on the protoplast release were tested in 10 mL of mixed enzyme solution (5 mg/mL of each enzyme). The conidia of *Aspergillus* sp. SCSIO SX7S7 were inoculated into 100 mL PDB medium and were grown at 28 °C for 12, 18, and 24 h. After the completion of the culture, the mycelia were collected and then digested by an enzyme mixture for 6, 12, and 18 h with 100 rpm at 28 °C. We found that 12 h mycelia culture combined with 18 h digestion is the optimal condition for the protoplast preparation of *Aspergillus* sp. SCSIO SX7S7 (Figure 2A,C and Appendix A). The above results indicated that the new mycelia, named germlings (germinated from the spore around 20–40 µm), produced the largest amount of fully digested protoplasts. Therefore, the optimal enzyme digestion time for the *Spiromastix* sp. SCSIO F190 strain was only tested using the germlings. The results showed that the optimal enzymolysis time of the *Spiromastix* sp. SCSIO F190 germlings for protoplast release is 9 h (Figure 2B,D and Appendix A).

Stable osmotic pressure is the critical factor for protoplasts to regenerate cell walls [31]. In this study, the osmotic pressure stabilizers 1 M sucrose and 1.2 M sorbitol were tested for protoplast regeneration in ANM regeneration medium with the optimized protoplast formation system. The regenerated number of diluted protoplasts was counted using microscopy. The results showed that the 1 M sucrose and 1.2 M sorbitol had similar effects and led to protoplast regeneration rates of nearly 45% and 40% for *Spiromastix* sp. SCSIO F190 and *Aspergillus* sp. SCSIO SX7S7, respectively (Appendix A). Taking the economy and efficiency into consideration, we found that 1 M sucrose is more suitable as the osmotic pressure stabilizer for the protoplast regeneration of both strains.

### 3.3. CRISPR-Cas9 Plasmid Construction for Gene Inactivation

The CRISPR-Cas9 technology has been seldomly applied to marine-derived fungi. To make use of this powerful technique in the marine-derived fungi *Spiromastix* sp. SCSIO F190 and *Aspergillus* sp. SCSIO SX7S7, we took advantage of two CRISPR-Cas9 editing systems with different sgRNA and Cas9 expression strategies, which were established by Liu et al. and Nodvig et al. in 2015, respectively [11,16,25], and constructed a new CRISPR-Cas9 vector pBSKII-to*Cas9-hph*-sgRNA. A plasmid pBSKII-to*Cas9*-*hph* [11,25] containing *T. reesei* codon-optimized *cas9* (to*Cas9*) and hygromycin B phosphotransferase (*hph*) coding sequences was used as the backbone. The sgRNA expression cassette including protospacer-sequence-flanked hammerhead (HH) and hepatitis delta virus ribozymes (HDV) ribozyme sequences, as well as a 6 bp inverted repeat of the 5-end of the protospacer to complete the hammerhead cleavage site under the control of *gpdA* promoter (P*gpdA*) and the *trpC* terminator (T*trpC*), was amplified from pFC330 [16] (Figure 3). The two ribozyme sequences ensure the liberation of the sgRNA from the transcript in the nucleus. The new CRISPR-Cas9 circular vector pBSKII-to*Cas9*-*hph*-sgRNA was constructed by inserting the amplified sgRNA expression fragments into the backbone vector via one-step isothermal assembly (Figure 3).

### 3.4. Target Gene Deletion in Spiromastix *sp*. SCSIO F190 and Aspergillus *sp*. SCSIO SX7S7

To test the functionality of our system, we first attempted to mutate the testing gene *creA* encoding a carbon catabolism repressor and protein kinase *cak1* in *Spiromastix* sp. SCSIO F190 and *Aspergillus* sp. SCSIO SX7S7, respectively. The successful mutagenesis of the two genes is easy to monitor, as the inactivity of these two genes usually results in a phenotype change with strong growth defects [32,33,34,35]. For the *Spiromastix* sp. SCSIO F190 strain, the CRISPR-Cas9 plasmids containing either no sgRNA or a sgRNA with a protospacer targeting the exon of *creA* were introduced into the protoplast of *Spiromastix* sp. SCSIO F190 via PEG. After 15 days of growth, we found around 28 out of 102 colonies with apparent phenotype changes when *creA* was targeted (Appendix A). Ten of those colonies with growth changes were picked out and their genomic DNA was purified. Then, the complete *creA* gene sequence was amplified using primers located in the 5′ and 3′ UTRs of the *creA* gene. Unexpectedly, half of the 10 transformants failed to show any PCR bands, and the remaining five colonies gave identical PCR products to the wild-type results (Figure 4A). However, all 10 transformants showed identical PCR products to the wild-type results when using a forward primer targeted to the sgRNA binding site and a reward primer located around 600 bp after the PAM sequence (Figure 4B), which suggested that the mutations did not happen in the sgRNA binding site as well as the subsequent sequence, while the above failed PCR results were not due to the DNA template quality but to large fragment deletion or insertion. To further examine any single base or only a few bases in mutagenesis that cannot be recognized from the DNA gel, the five PCR fragments encompassing the full *creA* gene were sequenced and two of them showed one base deletion in the sgRNA binding site (Figure 4C). For the other three clones with growth defects that failed to detect any mutations, we found two different phenotypes on non-selective medium after they had been growing for a long time (Appendix A), while the pure Δ*creA* mutant exhibited clear growth changes with less conidiation compared to the wild-type strain (Figure 4D and Appendix A), indicating that the original transformants were heterokaryotic with transformed–mutated and non-transformed–non-mutated nuclei.

Similarly, an sgRNA gene with a protospacer targeting the exon of *cak1* in the CRISPR-Cas9 expression vector was introduced into the protoplast of *Aspergillus* sp. SCSIO SX7S7 via PEG. Six transformants with serious growth defects (Figure 5E and Appendix A) were picked out and their genome DNA were extracted. Next, the DNA regions surrounding the target gene were amplified using primers flanking the *cak1* gene. Consistent with the mutation events in the *Spiromastix* sp. SCSIO F190 strain, five of the six transformants failed to show any PCR bands, suggesting that large unknown fragment deletion or insertion happened in most transformants (Figure 5A). The only clone (clone 5) that showed a normal PCR band similar to the wild-type strain was sequenced and revealed a 14 bp deletion around the PAM site (Figure 5D). Serendipitously, we obtained a non-specific band of clone 1 when using a forward primer targeted to the sgRNA binding site and reward primer located around 500 bp after the PAM sequence due to non-specific priming of the forward primer. The sequencing of this non-specific band demonstrated that the sequence near the sgRNA expression cassette in the pBSKII-to*Cas9*-*hph*-sgRNA plasmid was inserted into the cleavage site of Cas9 3 bp upstream of PAM (Figure 5D), confirming our above speculation that the failed PCR might have been caused by the insertion of a large fragment. To examine the insertion in further detail (e.g., how long of a sequence from the pBSKII-to*Cas9*-*hph*-sgRNA plasmid was inserted), forward primers located at different sites of the CRISPR-Cas9 vector combined with a reverse primer targeted to the *cak1* gene were used to amplify the junction sequence (Appendix A). As expected, different lengths of PCR products were obtained, except for a region with a length larger than 10 kb (Appendix A), which may have been due to the limitations of long-range PCR testing. Sequencing of the above PCR bands revealed that the contiguous vector was inserted at the target site. Additionally, PCR reactions using ITS1/4 primers with the above purified DNA samples were performed to further confirm that the failed PCR result was not due to the quality of the DNA template but rather to the unknown insertion or deletion (Figure 5C).

### 3.5. CRISPR-Cas9 Can Efficiently Mutate Histone Deacetylase Gene for Novel Natural Products Activation

The above results show that the established CRISPR-Cas9 system can efficiently induce gene disruption in both marine-derived strains. Hence, the CRISPR-Cas9 system was further applied to activate the silent gene clusters. The inactivation of histone deacetylases (HDACs) has been demonstrated to be an efficient approach for cryptic gene cluster activation [6,36,37,38]. In this study, *rpd3*, a gene encoding a classical histone deacetylase in *Aspergillus* sp. SCSIO SX7S7, was deleted, and the associated phenotypic and metabolic changes were evaluated. Three clones carrying potential *rpd3* mutations were picked out and confirmed via genotyping PCR (Appendix A). Similar to the above situations, two of the *rpd3* mutants failed to show any PCR bands using primers targeting different sites of the *rpd3* gene. The sequencing of an unspecific PCR band from clone 2 revealed that a part of the Cas9 coding sequence was inserted into the *rpd3* gene (Appendix A). Inconsistent with the above insertion, only a 1.2 kb PCR product was observed when using primers located at different sites of the transformed vector paired with a reverse primer targeted to the *rpd3* gene (Appendix A), suggesting that the insertion may be composed of rearranged segments of the transformation vector. The removal of *rpd3* resulted in slower growth and defective sporulation in *Aspergillus* sp. SCSIO SX7S7 (Appendix A). The metabolite extraction and HPLC analysis revealed the production of novel compounds in all three independent *rpd3* mutants compared to the wild-type strain (Figure 6). Moreover, the major secondary metabolites of the wild-type strain were significantly decreased in *rpd3* mutants (Figure 6), which suggested that the disruption of *rpd3* reduced the gene expression levels of compounds **1**–**12** associated biosynthetic gene clusters. In contrast, the gene clusters that are responsive to the biosynthesis of putative novel compounds were transcriptionally activated.

## 4. Discussion

For novel species, an efficient genetic manipulation system is important for the elucidation of the fungal molecular mechanisms behind the industrial applications and the discovery of natural products. CRISPR-Cas9-based technologies have reached almost every corner of the genetic manipulation field, providing powerful tools to edit the genomes of plants, animals, bacteria, fungi, and many other organisms. The CRISPR era of fungi began in 2015, when Zou and colleagues first demonstrated that CRISPR-Cas9 is an efficient gene disruption tool in the filamentous fungus *Trichoderma reesei* by adopting a “producing Cas9 in vivo while transcribing sgRNA in vitro” strategy [11]. In the same year, Mortensen’s team built a single-plasmid-based CRISPR-Cas9 genome editing toolkit for Aspergillus, producing both Cas9 and sgRNA in vivo [16]. Later, CRISPR-Cas9 was successfully used in many other filamentous fungi; for example, Hu and his lab members established an efficient CRISPR-Cas9-based gene disruption strategy via the simultaneous transformation of in vitro transcriptional gRNA and the linear maker gene cassette into Cas9-expressing fungi [25].

In the current study, we developed a new CRISPR-Cas9 and sgRNA expression plasmid pBSKII-to*Cas9*-*hph*-sgRNA by reassembling plasmid elements from previous studies based on the following considerations: (i) Transforming the linear resistance maker gene fragment into the Cas9-expressing fungi usually requires two selective markers available for the target strain; however, wild-type fungi from high-salt marine habitats have evolved significant resistance to multiple antibiotics [39,40], meaning most marine fungi, including *Spiromastix* sp. SCSIO F190, are not able to obtain two distinctive selective markers for the transformation of the *cas9*-expressing plasmid and linear maker gene fragment or sgRNA separately. (ii) Generally, the sgRNA can be expressed both in vitro and in vivo, but the in vitro transcription and purification process is laborious and costly. (iii) RNA polymerase type III promoters (U6 and U3 promoters) are frequently applied for the in vivo transcription of the sgRNA [41]. Unfortunately, these kinds of promoters are ill-defined in filamentous fungi [42]. Additionally, the CRISPR target sequences recognized by the U6 and U3 promoters are constrained with a certain sequence specificity [43]. Therefore, employing the ribozyme–sgRNA–ribozyme transcription cassette controlled by the RNA polymerase II promoters could guarantee the production of sgRNA in most species, as the functional sequences of RNA polymerase II promoters are well characterized in filamentous fungi. (iv) Even if the AMA1 sequence from *A. nidulans* permits autonomous plasmid replication, some integrations of the AMA1 vector into the genome have been observed in *Gibberella fujikuroi* [44]; thus, whether the AMA1 sequence can support episomal DNA delivery in other fungal species with the highly active non-homologous end-joining (NHEJ) repair pathway is still unclear. Moreover, adding the AMA1 sequence into the Cas9 and sgRNA expression vector would lead to a plasmid length larger than 15 kb, resulting in decreased isothermal assembly efficiency. Based on the above information, we tried a strategy involving the expression of both Cas9 and sgRNA in vivo in a single plasmid without the AMA1 sequence. Our results demonstrated that this system functions well in two phylogenetically distinct marine fungi. To the best of our knowledge, this is the first reported demonstration of genome editing in the *Spiromastix* genus of *Ascomycete* fungi.

Most studies have shown that CRISPR-Cas9-induced mutations are predominantly short deletions (1–400 bp) [13,42,45,46]; on the contrary, the most abundant modifications (more than 50% of tested strains) induced by CRISPR-Cas9 in the marine-derived fungi *Spiromastix* sp. SCSIO F190 and *Aspergillus* sp. SCSIO SX7S7 were large fragment insertions or deletions. Two occasional non-specific PCR events revealed that the insertion fragments come from the CRISPR-Cas9 plasmid that is broken at a random site, indicating the integration of the transforming plasmid at the Cas9 cut site through the NHEJ repair pathway. The high-frequency integration of the transforming construct at the Cas9 cut site has been also described in several other fungal species, e.g., *A. fumigatus* and *Sclerotinia sclerotiorum* [15,25,47], but not in all tested species, even when similar CRISPR-Cas9 plasmids were used [13]. The mechanism behind the species-specific mutation patten is still not well understood. Recently, Rollins et al. showed that the transformed plasmid can express Cas9 and sgRNA quickly before being inserted itself into the target sgRNA binding site in the filamentous pathogen *Sclerotinia sclerotiorum* [13]. Nevertheless, whether the pattern of plasmid insertion in the two marine fungal species is the same as for *Sclerotinia sclerotiorum* still needs to be further confirmed by investigating the copy number of the CRISPR-Cas9 expression plasmid in the host strain.

An important caveat when using CRISPR-Cas9 technology is the potential for off-target effects. Nonetheless, assessments of the probability of off-target mutations in different filamentous fungal species through whole-genome sequencing have revealed that the off-target effects after CRISPR-Cas9 mutagenesis may not be a major issue [42,47]. In the current study, we also tried to limit the off-target effects by improving the specificity of the sgRNA, in which the 13 bp sequence adjacent to the PAM is unique in the genome, as Cas9 or sgRNA does not recognize and edit DNA sites with any number of mismatches (within 10–12 bp) near the PAM [42,48,49]. In addition, the consistency of the induced novel natural products from three independent *rpd3* mutants also suggested a low probability of off-target effects. However, it is still advisable to keep the Cas9 and sgRNA expression at a minimum level to mitigate this probability. The transient expression CRISPR-Cas9 system and plasmid-free CRISPR-Cas9 system assembling the Cas9 protein and sgRNA to form a stable RNP in vitro will be further tested to mitigate the risk of off-target effects.

The implementation of the CRISPR-Cas9 system presented here is expected to lead to an acceleration in the discovery and activation of novel SMs, especially for those non-domesticated fungal species. The fungal secondary metabolite gene clusters are controlled by a complex regulatory network involving multiple proteins and complexes that not only respond to various environmental stimuli, but also regulate the cellular chromatin modifications [50]. The CRISPR-Cas9 toolkit developed in this study provided a powerful platform for the genetic manipulation of these regulators, inducing the production of novel natural products. Using a single CRISPR-Cas9 plasmid, we successfully mutated a histone deacetylase gene, *rpd3*, and activated a series of novel compounds that are not produced in the wild-type strain.

Taken together, in this study an efficient and robust genetic manipulation system was developed in the marine-derived fungi *Spiromastix* sp. SCSIO F190 and *Aspergillus* sp. SCSIO SX7S7. Using this approach, several novel secondary metabolites were activated in the *rpd3* mutant of *Aspergillus* sp. SCSIO SX7S7. The CRISPR-Cas9 toolbox could expedite the discovery and biosynthetic mechanism elucidation of increasingly invaluable natural products in *Spiromastix* sp. SCSIO F190 and *Aspergillus* sp. SCSIO SX7S7, as well as many other marine-derived filamentous fungi.

## Figures and Tables

**Figure 1 jof-08-00715-f001:**
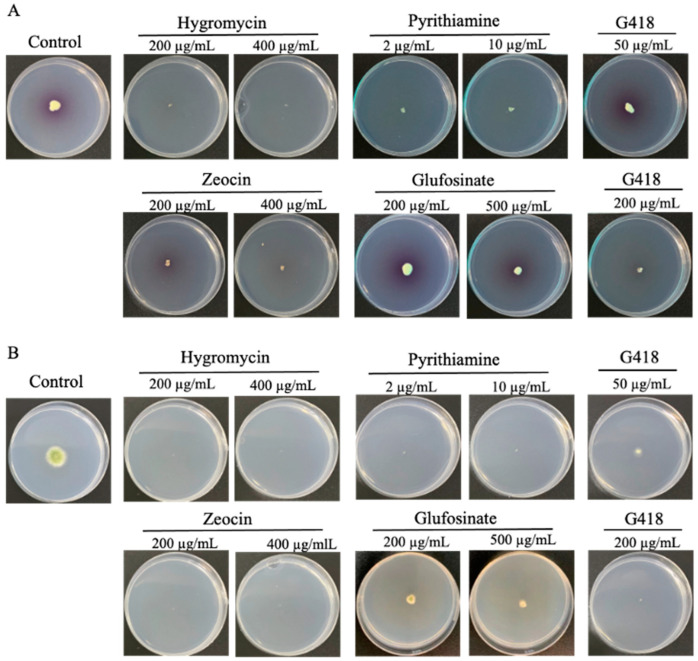
The sensitivity of the wild-type strains *Spiromastix* sp. SCSIO F190 (**A**) and *Aspergillus* sp. SCSIO SX7S7 (**B**) in the ANM plate with different concentrations of hygromycin B, pyrithiamine, zeocin, glufosinate, and G418 at 28 °C for 8 and 4 days, respectively. Control: culture without the addition of antibiotics.

**Figure 2 jof-08-00715-f002:**
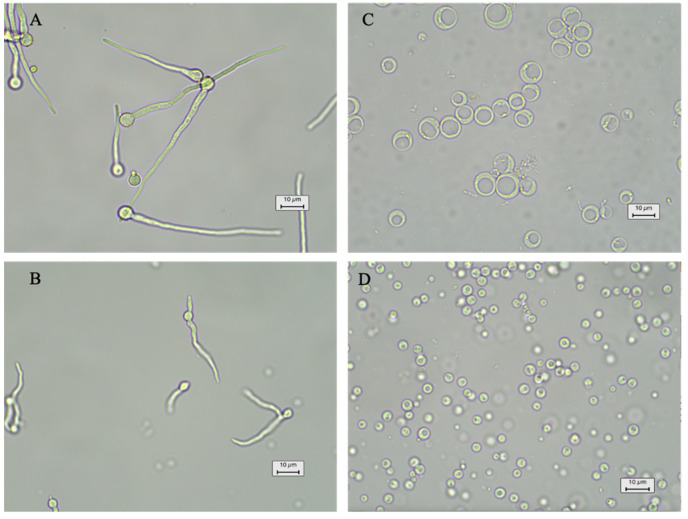
A microscopic check of germination and protoplasts released from the mycelia: (**A**,**B**) germlings of *Aspergillus* sp. SCSIO SX7S7 (**A**) and *Spiromastix* sp. SCSIO F190 (**B**) under 12 h and 24 h culture, respectively; (**C**,**D**) protoplasts of *Aspergillus* sp. SCSIO SX7S7 (**C**) and *Spiromastix* sp. SCSIO F190 (**D**) after 18 h and 9 h digestion, respectively.

**Figure 3 jof-08-00715-f003:**
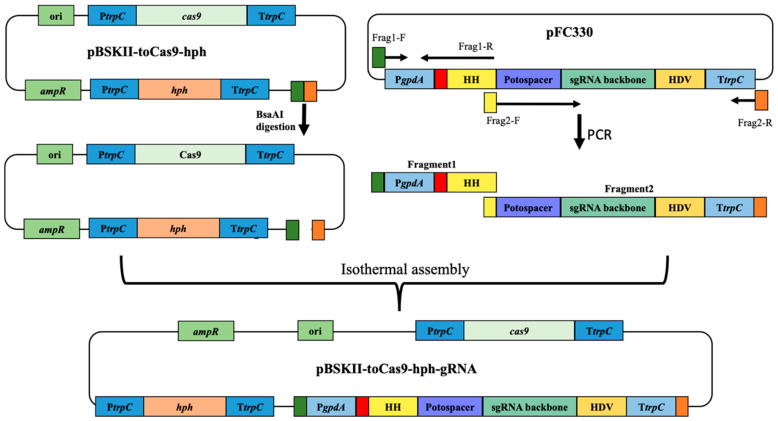
The construction of new CRISPR-Cas9 vectors (pBSKII-to*Cas9*-*hph*-gRNA) for the directed mutagenesis of marine-derived fungi. The vector backbone for the construction of new fungal vectors was derived from the plasmid pBSKII-to*Cas9-hph*, and was digested by the BsaAI restriction enzyme. Variable sgRNA genes controlled by the *gpdA* promoter and *trpC* terminator were amplified from the pFC330 plasmid using two pairs of primers. The Fragment1-containing *gpdA* promoter sequence, a 6 bp inverted repeat of the 5′-end of the protospacer (in red), the hammerhead (HH) ribozyme sequence, and 5′-end sequence complementary (in dark green) to the left side of BsaAI cutting site in plasmid pBSKII-to*Cas9*-*hph* was amplified by primers Frag1 F/R. Fragment2 carrying the protospacer sequence, the sgRNA backbone sequence, the hepatitis delta virus (HDV) ribozyme sequence, and the *trpC* terminator sequence flanked by the 15 bp complementary sequence to fragment1 (in yellow) and 20 bp complementary sequence to the right side of BsaAI cutting site in plasmid pBSKII-to*Cas9*-*hph* was amplified by the primer Frag2 F/R. The resultant PCR products of flanking regions and the BsaAI-digested pBSKII-to*Cas9*-*hph* plasmid can be joined together to form a new construct using the isothermal assembly method. For simplicity, all complementary ends are visualized in the same color and no DNA elements in the above figure are drawn to scale.

**Figure 4 jof-08-00715-f004:**
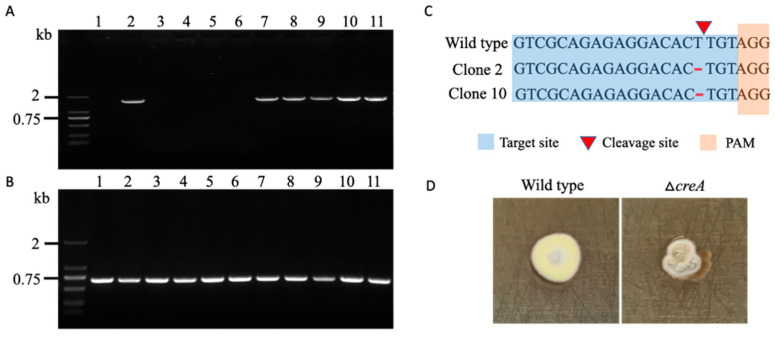
The inactivation of the *creA* gene in *Spiromastix* sp. SCSIO F190 using the CRISPR-Cas9 system: (**A**) DNA amplification of the DNA regions surrounding the full *creA* genes in the ten clones picked out from the regeneration plate using primers flanking the *creA* genes; (**B**) DNA amplification of the DNA regions surrounding the PAM sites of the ten clones picked out from the regeneration plate using primers flanking the PAM sites; (**C**) sequence analysis of the PCR products generated in (**A**); (**D**) morphological comparison between the Δ*creA* and F190 wild-type strains on a complete plate at 28 °C for 14 days ((**A**,**B**) lane 1–10: tested *creA* mutated clones; lane 11: wild-type strain).

**Figure 5 jof-08-00715-f005:**
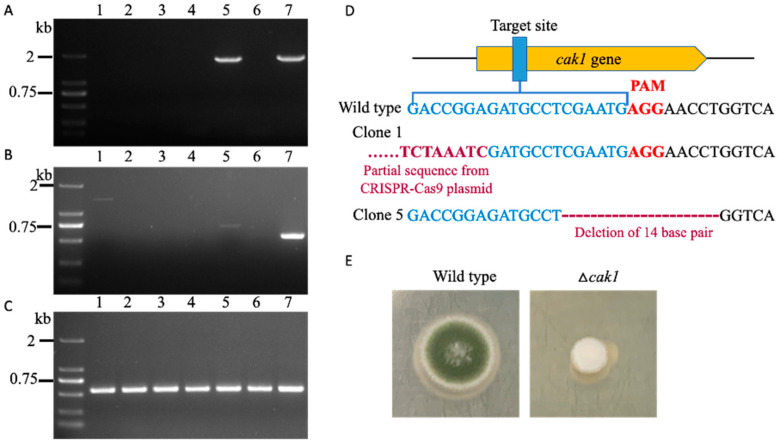
The inactivation of the *cak1* gene in *Aspergillus* sp. SCSIO SX7S7 using the CRISPR-Cas9 system: (**A**) DNA amplification of the DNA regions surrounding the full *cak1* genes of the six clones picked out from the regeneration plate using primers flanking the *cak1* genes; (**B**) DNA amplification of the DNA regions surrounding the PAM sites of the six clones picked out from the regeneration plate using primers flanking the PAM sites; (**C**) positive control used for checking the DNA template quality of the six clones picked out from the regeneration plate using primers flanking the ITS region; (**D**) sequence analysis of PCR products of clones 1 and 5; (**E**) morphological comparison between the representative Δ*cak1* mutant and 7S7 wild-type strains on a complete plate at 28 °C for 4 days ((**A**–**C**) lane 1–6: tested *cak1* mutation clones; lane 7: wild-type strain).

**Figure 6 jof-08-00715-f006:**
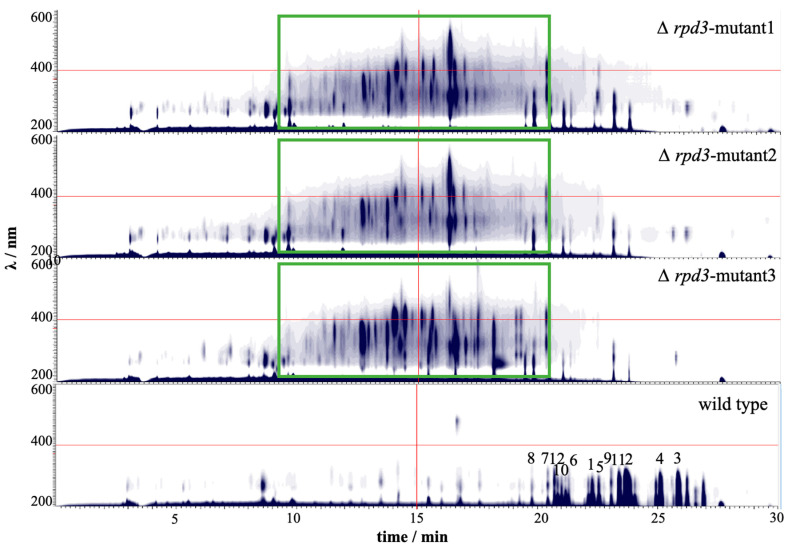
HPLC-DAD contour plot of extracts from the wild-type *Aspergillus* sp. SCSIO SX7S7 and three independent transformants of Δ*rpd3* mutants. The boxed region shows the novel compounds induced in Δ*rpd3* mutant strains. The red lines of each figure indicates the middle position of x and y axis for better comparision. The number indicates the compounds isolated from the wild-type strain ((**1**) asperdepside A; (**2**) nornidulin; (**3**) nidulin; (**4**) aspergillusidone B; (**5**) 2,7-dichlorounguinol; (**6**) aspergillusidone C; (**7**) aspergillusidone A; (**8**) 2-chlorounguinol; (**9**) emeguisin A; (**10**) 7-carboxyfolipastatin; (**11**) agonodepside A; (**12**) unguidepside A).

## Data Availability

The data presented in this study are available in this manuscript, and constructs can be requested from the corresponding author.

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
