# Peer review of "Development of the CRISPR-Cas9 System for the Marine-Derived Fungi Spiromastix sp. SCSIO F190 and Aspergillus sp. SCSIO SX7S7"

_jof, 2022, doi:10.3390/jof8070715_

Round 1

Reviewer 1 Report

The manuscript by Chen et al. describes the development of gene editing for two marine-derived fungi through the CRISPR-Cas9 system. Besides, the basic techniques required for genetic manipulation in both fungal strains were also established, such as resistance gene marker, protoplast preparation, transformation system, and other genetic toolboxes. The presented data are interesting and scientifically sound. While the paper contains enough good quality to warrant publication in JoF, however, there are some things to clarify and modify the text as follows:

-          - Did the authors measure the percentage of efficiency in gene editing using the developed CRISPR-Cas9 system in both fungal strains? The authors mentioned only the number of transformants and showed the representative colonies for each strain.

-          - In Figure 6, please explain why the secondary metabolites (compounds 1-12) presented in the wild type disappeared by the rpd3 disruption. Did the authors identify the major novel compounds generated by the rpd3 disruptant strain?

-          In Figures S5 and S6, the fungal names should be defined in the figure legends.

-          In the legend of Figure S8 and Table S1, do the authors know the species name; Aspergillus unguis? It was not mentioned elsewhere in the manuscript.

-          Please change the words “PCR amplification” to “DNA amplification” or “PCR analysis” throughout the manuscript and supplementary file.

-          There are some grammatical errors and mistyping. Please check English the manuscript thoroughly.

Author Response

Dear editor and reviewers:

Thank you very much for your suggestions and comments , all questions have been answered point by point, and the manuscript has been revised accordingly.

Author Response

(The authors gave the same response as above.)

Reviewer 3 Report

The research and experimental design were fine, and the research was of interest to the community. The figures were well prepared. The English writing need some help from a native English speaker. Attached please find annotated manuscript with some writing issues highlighted. 

Author Response

Dear editor and reviewers:

Thank you very much for your  comments, the highlight writing issues have been addressed one by one , and the english writing has been checked and revised with help from a native English speaker Vikram Norton.
